# Multi-Section Traffic Flow Prediction Based on MLR-LSTM Neural Network

**DOI:** 10.3390/s22197517

**Published:** 2022-10-04

**Authors:** Ruizhe Shi, Lijing Du

**Affiliations:** School of Safety Science and Emergency Management, Wuhan University of Technology, Wuhan 430079, China

**Keywords:** time series, traffic prediction, long short-term memory, multiple linear regression

## Abstract

As the aggravation of road congestion leads to frequent traffic crashes, it is necessary to relieve traffic pressure through traffic flow prediction. As well, the traffic flow of the target road section to be predicted is also closely related to the adjacent road sections. Therefore, in this paper, a prediction method based on the combination of multiple linear regression and Long-Short-Term Memory (MLR-LSTM) is proposed, which uses the incomplete traffic flow data in the past period of time of the target prediction section and the continuous and complete traffic flow data in the past period of time of each adjacent section to jointly predict the traffic flow changes of the target section in a short time. The accurate prediction of future traffic flow changes can be solved based on the model supposed when the traffic flow data of the target road section is partially missing in the past period of time. The accuracy of the prediction results is the same as that of the current mainstream prediction results based on continuous and non-missing target link flow data. Meanwhile, there is a small-scale improvement when the data time interval is short enough. In the case of frequent maintenance of cameras in actual traffic sections, the proposed prediction method is more feasible and can be widely used.

## 1. Introduction

In recent years, with the continuous increase of automobile ownership, there are approximately 850 million automobiles in the world. However, the carrying capacity of the road is inconsistent with the growth of car numbers, resulting in long-term congestion and stagnation on the road, which not only reduces traffic efficiency and increases residents’ travel time but also increases the risk of traffic crash. In order to ensure the smoothness of the road, it is significant to accurately predict the traffic flow of each section and provide suggestions for traffic diversion. In this paper, we define traffic flow as the number of vehicles passing in a specified time in the road slice monitored by the camera. Traffic flow data of each adjacent road section should be combined to jointly predict the traffic flow of the target road section allowing partial missing of the past flow data in this paper.

Traffic flow prediction is regarded as an important sub-topic in transportation. The prediction methods used in traffic flow have been updated with the development of the times. With the application of big data technology in the field of traffic flow, the neural network method has been widely used in traffic flow prediction. However, this prediction method has strict requirements on data. If the data is partially missing, the effect will be affected. In addition, the traffic flow data changes greatly in the short-term, has many uncertainties, and shows periodicity relative to time. In view of the fact that the current research in this area is not rich enough, the existing methods have a large amount of calculation, high data requirements, and a need for complete and correct data sets, we propose a method to obtain accurate prediction when the traffic flow data of the target road section is incomplete. We use the cyclic neural network to predict according to the periodicity of the data, and obtain the prediction value with high accuracy of the target road section through the linear regression of the traffic flow data of multiple road sections.

Today’s methods are based on the coordinate data of each road section combined with the graph neural network, and the traffic flow prediction of multiple road sections is obtained after the convolution operation. This method requires a high accuracy of data. If the data is wrong, the whole model result will be inaccurate. The MLR-LSTM proposed by us has a high fault tolerance rate. The combination of traditional statistical methods and deep learning makes the model easier to explain and understand.

In particle situations, the cameras on the road sections will lose traffic flow data in some periods of time due to warranty and other reasons, so the historical traffic flow data of the target road sections are likely to be incomplete. The method we propose can effectively avoid the impact of this situation on the prediction results, which is the purpose of our research.

This study contributes in three ways. First of all, this paper combines the spatiotemporal information of traffic flow data, interpolates, and supplements the data of adjacent road sections under the same time slice, and then applies multiple linear regression to predict the missing road section data. A method of data complement in each time slice is applied to traffic flow data. Secondly, the article focuses on the combination of traditional statistical methods and neural networks, which are well-applied in traffic flow prediction. Through the combination of the traditional measurement model and the network model, the interpretability of the model is enhanced. Finally, the article simplifies the construction of the model and relaxes strict requirements for data integrity, so that the model is both effective and widely accepted. Only the historical traffic flow data of each adjacent road section is needed to predict when part of the historical data of the target road section is missing.

The rest of this paper is organized as follows. Section 2 reviews the classic models of traffic flow prediction and shows the literature positioning of this study. Section 3.1 introduces the framework and process of this model. Section 3.2 introduces and improves the long and short cycle neural network. The model is built in Section 3.4. In Section 4, comparative experiments and an analysis of experimental results are carried out. Section 5 concludes the evaluation of our model and discusses future research directions.

## 2. Literature Review

Analyzing the time series data of traffic flow, mastering the rules, and predicting the flow change in the future for a period of time can provide better traffic control decisions for traffic managers. Therefore, the research on traffic flow prediction has become richer and richer and has kept pace with the times since the last century. Therefore, in addition to the expansion of various branches of traffic flow forecasting, such as single-point forecasting and multi-point forecasting, single-period forecasting and multi-period forecasting, a huge treasure has been formed in the change of forecasting methods. At present, traffic flow prediction methods are mainly divided into model-driven and data-driven methods.

Model-driven traffic flow prediction methods include various prediction models based on nonlinear theory. The nonlinear theories and methods used mainly include catastrophe theory [1], chaos theory [2], wavelet analysis [3], and so on. This kind of model can better fit the characteristics of multimodality, mutation, inaccessibility, divergence, and lag of traffic flow state. Among them, Huang Yanguo et al. [1] (2022) proposed a traffic flow cusp catastrophe model based on traffic wave theory with traffic density as state variables and traffic flow and wave speed as control variables. Anyu Cheng et al. [2] (2017) used the maximum Lyapunov exponent to identify the chaotic characteristics of traffic flow related to speed, occupancy, and flow, and produced a traffic flow prediction algorithm based on multi-source and multi-measure. Yanchi Li et al. [3] (2020) improved a prediction model by combining wavelet analysis and neural networks, which improved the prediction accuracy through the combination of wavelet denoising and BP neural network.

With the rapid development of big data technology, data-driven methods have been widely studied. The early research on traffic flow prediction mainly adopts traditional statistical models, such as the historical average model, time series analysis model, Kalman filter analysis model, support vector regression model, and so on. The typical ha averages the data throughout the period and takes the average value as the prediction value, but this method has low prediction accuracy and the prediction result of traffic flow is not ideal. Time series analysis models mainly include moving average model, autoregressive moving average model, integrated moving average autoregressive model, and so on. Dharyll Prince m et al. [4] (2019) developed an autorepressant integrated moving average (ARIMA) model to analyze the traffic flow dynamics of the Philippines. Sheng Yang Ge et al. [5] (2013) drew an exponential smoothing and trend moving average method. Lingru Cai et al. [6] (2021) introduced the maximum correlation entropy to deduce the Kalman filter to formulate the traffic flow prediction task as well as achieved superior performance. Zhao Liu et al. [7] (2018) introduced a combination of K—nearest neighbor and support vector regression to improve the accuracy. However, in most cases, the traditional statistical model has certain requirements or assumptions for the data, and requires the model itself to have a relatively clear mathematical form. However, in most cases, people are usually unable to make any assumptions about the distribution of data in the real world.

In recent years, with the extensive application of artificial intelligence methods such as machine learning and deep learning in the field of transportation, the prediction of traffic flow has achieved good research results in terms of a nonlinear relationship. Machine learning methods [8] (2022) can analyze complex and diverse data in-depth without any assumptions about the data. Understanding how to deeply analyze complex and diverse data through machine learning and make efficient use of information has become one of the main problems paid attention to by big data. Pan Chengsheng et al. [9] (2022) used the neutral net to achieve traffic prediction. Meanwhile, Lin Guancen et al. [10] (2022) succeeded in traffic prediction based on the traditional machine learning method. As well, Yuen Man Chung et al. [11] (2022) used a competition mechanism multi-objective particle swarm optimization algorithm to solve the traffic flow problem efficiently.

The most important data information is the information based on the time dimension and the space dimension. The traffic flow prediction method based on these two dimensions is deeply studied. Wang Jun et al. [12] (2022) developed a method of automatically obtaining spatial dependence in data, which can automatically obtain the spatial state and spatial dependence using a multi graph advantageous neural network to predict traffic flow in time and space. V é lezserrano Daniel et al. [13] (2021) performed a short-term prediction of traffic flow in Madrid, using different types of neural network architectures with a focus on convolutional residual neural networks. Xinyu Chen et al. [14] (2021) obtained better prediction results after a Bayesian decomposition of multidimensional data. Peixiao Wang et al. [15] (2022) proposed a multi-view bidirectional spatio-temporal network based on the spatio-temporal network. Shaokun Zhang et al. [16] (2022) proposed a graph-based multi-sensor prediction framework which improved the accuracy of prediction

In neural network prediction, long short term memory (LSTM) is widely used in various models, and has achieved good results. Wangyang Wei et al. [17] (2019) realized traffic flow based on AutoEncoder and LSTM. Ali Ahmad et al. [18] (2021) advised a unified dynamic deep spatial temporary neural network model based on progressive neural networks and long short-term memory to simultaneously predict crowd flows in every region of a city. Alkhede et al. [19] (2021) selected three machine learning approaches namely fuzzy logic, long short term memory (LSTM), and decision trees to predict traffic flow. The results show that LSTM has proven to have the best results of the three models. Wang Ke et al. [20] (2021) put forward a short-term traffic flow prediction model based on the attention mechanism and the 1dcnn-lstm network

The model-driven method [21] (2020) is used to build a model based on the understanding of the traffic model, but its accuracy and applicability are limited due to the complexity of the actual traffic environment. The data-driven method focuses on the mapping relationship between data and phenomena, but the demand for data is large, and the understanding and application depth of traffic mechanisms is insufficient. Therefore, this paper draws a method based on the combination of multiple linear regression and LSTM. The operation law of intersection traffic is obtained by multiple linear regression on the data of relevant intersections, and predicted in combination with LSTM. Compared with the spatiotemporal graph convolution network, the amount of data is greatly reduced, and the model is simplified when the accuracy is not much different.

## 3. Methodology

### 3.1. Multi Intersection Traffic Flow Prediction Framework

The framework of the proposed multi-intersection traffic prediction method is described in Figure 1.

First, we preprocess data by data cleaning. Then, the data of adjacent sections of the target section are trained based on LSTM, in which the training set, test set, and inspection set are set to train the parameters in the LSTM network, predict each adjacent section and evaluate the prediction results. After deep learning, multiple linear regression is carried out between the adjacent road section and the target road section to obtain the regression parameters. Finally, combining the regression parameters with the prediction results of LSTM for each adjacent road section, the prediction results for the target road section based on the proposed new method are calculated.

Part 1: The multi-dimensional data is completed with cubic Hermite interpolation, so that the data under each time slice is complete and the outliers are eliminated.

Part 2: The LSTM network is established, the LSTM network with existing road section data is trained, and an optimal network structure is obtained through parameter adjustment.

Part 3: The data of adjacent road sections are used to conduct multiple linear regression for the road sections to be tested, and the correlation coefficient between them is obtained.

Part 4: The adjacent data predicted by the LSTM network is combined with the correlation coefficient obtained by multiple linear regression to obtain the traffic flow of the target section.

### 3.2. LSTM Building Module

#### 3.2.1. Building Internal Structure of the Cells

The long-term and short-term memory model (LSTM) is a neural network to improve the structure of the Recurrent Neural Network (RNN). The main purpose of this model is to solve the problems of gradient disappearance and explosion in the process of long sequence training, and make the cyclic neural network have stronger and better memory performance. In short, LSTM can perform better in longer sequences than RNN. The ability of longer dependencies is conducive to dealing with events with longer intervals

Compared with ordinary time series analysis such as RNN, LSTM adds a forgetting gate, memory gate, and output gate to avoid gradient explosion or disappearance. The three gates represent three stages within the LSTM: forgetting, selecting memory, and output. These three stages are used to modify the information in the cell state.

Compared with RNN, LSTM as a whole not only flows with time, but also cell state flows with time. Cell state represents long-term memory.

In Figure 2, the internal structure of the LSTM is reflected, and the internal structures are described below

##### Forgetting Stage

This stage is mainly used to selectively forget the input from the previous node and determine what information we discard from the cellular state. The position and update formula of forgetting gate is as follows:(1)$$f_{t}=\sigma \left(W_{f}\cdot \left[h_{t-1},x_{t}\right]+b_{f}\right)$$

Among Equation (1), $f_{t}$ is the weight matrix multiplied by $h_{t-1}$ and $x_{t}$ splicing vectors, and then converted to a value between 0 and 1 through a sigmoid activation function, which is used as a gating state.

##### Selecting Memory Stage

This stage is mainly to selectively remember the input in order to determine what new information is stored in the cell state. The position and structure formulas of the updating gate are as follows:(2)$$i_{t}=\sigma \left(W_{i}\cdot \left[h_{t-1},x_{t}\right]+b_{i}\right)$$
(3)$${\tilde{C}}_{t}=\tan h\left(W_{C}\cdot \left[h_{t-1},x_{t}\right]+b_{C}\right)$$

Then LSTM combines the forgetting gate, updating gate, upper layer memory cell value and memory cell candidate value to jointly determine and update the current cell state. The formula is as follows:(4)$$C_{t}=f_{t}\times C_{t-1}+i_{t}\times {\tilde{C}}_{t}$$

##### Output Stage

In the current state, this phase will determine which outputs will be considered. This output will be determined by our cell state. LSTM includes a separate output gate to realize the function. Its position and calculation formula are as follows:(5)$$O_{t}=(W_{o}[h_{t}-1,x_{t}]+b_{o})\;h_{t}=O_{t}*\tanh(C_{t})$$

#### 3.2.2. Establishing the Links of Cells

In this study, we build a double-layer LSTM model and add a dropout function to prevent overfitting as seen in Figure 3. A dropout function is performed when information is transferred between multi-layer cells at the same time. The dropout function is passed between cells. The horizontal direction is the LSTM calculated horizontally, so as to adjust the influence of the previous state on the current cell state.

#### 3.2.3. Definition of the Loss Function

It is necessary to define a loss function in the neural network so that the neural network can be adjusted to achieve more ideal results. This paper uses *MSE* as the loss function of the model. For the definition of *MSE*, we refer to the following formula., where f(x) is the target value and y is the predicted value.


(6)
$$\mathrm{MSE}=\frac{{\sum_{i=1}^{n}(f(x)-y)}^{2}}{n}$$


The function curve of *MSE* is smooth, continuous, and derivable everywhere. It is convenient to use a gradient descent algorithm, and is a commonly used loss function. Moreover, as the error decreases, the gradient also decreases, which is conducive to convergence. Even if a fixed learning rate is used, it can converge to the minimum quickly.

### 3.3. Data Preprocessing Module

#### Data Cleaning

Data cleaning is a critical step in data preprocessing that involves re-examining and verifying the data. The goal is to remove duplicate data, correct existing errors, and ensure data consistency. The data collected by the British highway detector is the source of this paper’s data. There is no duplicate data in this data source because the detector records every 15 min. In this paper, data cleaning is primarily concerned with incomplete data.

For the missing data at the same time interval, cubic Hermite interpolation is used.

It is required that the value of the interpolation polynomial on each interpolation node is equal to the value of the interpolated function, and its derivative on the inter-polation node is equal to the derivative of the interpolated function, that is, the fol-lowing formula:$$\{\begin{array}{c} H_{2n+1}\left(t_{i})=v(t_{i}\right)\\ H_{2n+1}^{\prime }\left(t_{i})=v\prime (t_{i}\right) \end{array}$$

In this formula, v is the traffic flow data about time *t*, and H is the data after interpolation. The interpolation interval in this article is 15 min. The v in the follow-ing is considered as the data that has been completed by interpolation.

After data cleaning, the deleted data accounts for less than 1% of the original data set, and the repaired data meets the experimental data quality requirements, which improves data quality by ensuring the integrity of data information.

### 3.4. Model Building

#### 3.4.1. Construction of Supervised Learning Training Model

The traffic flow information of different sections on the expressway is a time series, and there is a corresponding observation value at each time. The time series of adjacent different sections in the experimental data can be expressed as follows:$$\mathrm{data}\;=\left[\begin{array}{cccc} v_{1}\left(t_{1}\right) & v_{2}\left(t_{1}\right) & \cdots & v_{m+1}\left(t_{1}\right)\\ v_{1}\left(t_{2}\right) & v_{2}\left(t_{2}\right) & \cdots & v_{m+1}\left(t_{2}\right)\\ \vdots & \vdots & \cdots & \vdots \\ v_{1}\left(t_{n}\right) & v_{2}\left(t_{n}\right) & \cdots & v_{m+1}\left(t_{n}\right) \end{array}\right]$$
where *m* represents the number of monitored road sections which is *m*, number the monitored road sections according to 1, 2,…, *m*, *m* + 1. $v_{1}$, $v_{2}$, …, $v_{m+1}$ represent the traffic flow of road sections 1, 2,…, *m*, *m* + 1, respectively, and $t_{1},\;t_{2},t_{n}$ represent the corresponding observation time.

The observation data of the k-th observation intersection is shown in the following formula:$$\left[\begin{array}{c} v_{k}\left(t_{1}\right)\\ v_{k}\left(t_{2}\right)\\ \vdots \\ v_{k}\left(t_{n}\right) \end{array}\right]\left[\begin{array}{c} v_{1}^{k}\\ v_{2}^{k}\\ \vdots \\ v_{n}^{k} \end{array}\right]$$

Take the *k*-th intersection as an example to train the LSTM model. Before training the model, the sliding window is used to realize the sliding of the window through the shift function, and the experimental data is constructed into a supervised learning sequence:$$X_{k}=\left[\begin{array}{cccc} V_{1}^{k} & V_{2}^{k} & \cdots & V_{w}^{k}\\ V_{2}^{k} & V_{3}^{k} & \cdots & V_{w+1}^{k}\\ \vdots & \vdots & \cdots & \vdots \\ V_{n}^{k} & V_{n+1}^{k} & \cdots & V_{n+w}^{k} \end{array}\right]=\left[\begin{array}{c} X_{1}^{k,w}\\ X_{2}^{k,w}\\ \vdots \\ X_{n}^{k,w} \end{array}\right]\;Y_{k}=\left[\begin{array}{c} Y_{w+1}^{k}\\ Y_{w+2}^{k}\\ \vdots \\ Y_{w+n}^{k} \end{array}\right]$$
where, w represents the length of time window, i.e., the time step. *X* represents the training set *X*, which is a matrix of n*w. *Y* represents the training set *Y*. We predict the data of the next time period through w data, and recur the process in turn. Terminally, a total of n data is predicted.

The traffic flow of the k-th road is predicted from (n+w+1) to (n+w+1+p) through the trained LSTM. The prediction results are shown in the following formula:

And the total prediction matrix for m road sections is as shown in the following formula:$${\hat{L}}_{k}=\left[\begin{array}{c} {\hat{V}}_{n+w+1}^{k}\\ {\hat{V}}_{n+w+2}^{k}\\ \vdots \\ {\hat{V}}_{n+w+p+1}^{k} \end{array}\right]$$

And the total prediction matrix for m road sections is as shown in the following formula:$$\hat{L}=\left[\begin{array}{c} {\hat{L}}_{1}\\ {\hat{L}}_{2}\\ \vdots \\ {\hat{L}}_{m} \end{array}\right]$$

#### 3.4.2. Construction of Multiple Linear Regression Model

The change of traffic flow at an intersection is often affected by the change of traffic flow at several adjacent intersections. One variable is closely related to multiple variables. Therefore, a multiple linear regression model is constructed. When the target road section is the traffic flow change of the m-th road section, let $y=v_{m}$, $x_{i}=v_{i}$, and take the traffic flow information of the previous m-1 road section as the independent variable to establish the regression model as follows:(7)$$\{\begin{array}{l} y=\beta_{0}+\beta_{1}x_{1}+\beta_{2}x_{2}+\cdots +\beta_{m-1}x_{m-1}+\varepsilon \\ \varepsilon ∼N\left(0,\sigma^{2}\right) \end{array}$$

Equation (6), $\beta_{0},\beta_{1},\cdots ,\beta_{m},\sigma^{2}$ are the unknown parameters irrelevant to $x_{1},x,\cdots ,x_{m}$.

$\beta_{0},\beta_{1},\beta_{2},\cdots ,\beta_{m},\sigma^{2}$ are called the regression coefficient of the model, which is used to describe the change degree of dependent variable caused by the change of an independent variable, among them $\beta_{0}$ is a constant term.

Now n independent observation data $\left(y_{i},X_{i1},X_{i2},\cdots ,X_{im}\right)$ are obtained, and the following formula is inferred: (8)$$\{\begin{array}{l} y=\beta_{0}+\beta_{1}x_{i1}+\beta_{2}x_{i2}+\cdots +\beta_{m}x_{im}+\varepsilon \\ \varepsilon ∼N\left(0,\sigma^{2}\right) \end{array}$$

Record it as
(9)$$\begin{array}{c} X=\left[\begin{array}{cccc} 1 & x_{11} & \cdots & x_{1m}\\ 1 & x_{21} & \cdots & x_{2m}\\ \vdots & \vdots & \ddots & \vdots \\ 1 & x_{n1} & \cdots & x_{nm} \end{array}\right],Y=\left[\begin{array}{c} y_{1}\\ y_{2}\\ \vdots \\ y_{n} \end{array}\right],\\ \varepsilon ={\left[\varepsilon_{1},\varepsilon_{2},\cdots ,\varepsilon_{n}\right]}^{T},\beta ={\left[\beta_{0},\beta_{1},\beta_{2},\cdots ,\beta_{m}\right]}^{T} \end{array}$$

Then the original formula can be expressed as
(10)$$\{\begin{array}{c} Y=X\beta +\varepsilon \\ \varepsilon ∼N\left(0,\sigma^{2}E_{n}\right) \end{array}$$

Model parameter estimation
(11)$$Q=\overset{n}{\underset{i=1}{\Sigma }}\varepsilon_{i}^{2}=\overset{n}{\underset{i=1}{\Sigma }}{\left(b_{i}-\beta_{0}-\beta_{1}a_{i1}-\beta_{2}a_{i2}-\cdots -\beta_{m}a_{im}\right)}^{2}\;\frac{\partial Q}{\partial \beta_{j}}=0,j=0,1,2,\cdots ,n$$
(12)$$\left\{ \begin{array}{c} \frac{\partial Q}{\partial \beta_{0}}=-2\overset{n}{\underset{i=1}{\Sigma }}(b_{i}-\beta_{0}-\beta_{1}a_{i1}-\beta_{2}a_{i2}-\cdots -\beta_{m}a_{in})=0\\ \frac{\partial Q}{\partial \beta_{0}}=-2\overset{n}{\underset{i=1}{\Sigma }}(b_{i}-\beta_{0}-\beta_{1}a_{i1}-\beta_{2}a_{i2}-\cdots -\beta_{m}a_{im})a_{ij}=0 \end{array}\right.\;\hat{\beta }={\left(X^{T}X\right)}^{-1}X^{T}Y$$
(13)$$\hat{y}={\hat{\beta }}_{0}+{\hat{\beta }}_{1}x_{1}+{\hat{\beta }}_{2}x_{2}+\cdots +{\hat{\beta }}_{m}x_{m}$$
where $\beta_{0},\beta_{1},\cdots ,\beta_{m}$ is the estimated value of regression parameters.

#### 3.4.3. Multiple Linear Regression Model Test

##### Goodness of Fit Test

Let the mean of *y* be $\bar{y}$ the fitting value of *Y* is $\hat{y}$, the formula is shown in the figure below:(14)$$R^{2}=\frac{\mathrm{SSR}}{\mathrm{SST}}=\frac{\Sigma_{i=1}^{n}{\left(\hat{y}-\bar{y}\right)}^{2}}{\Sigma_{i=1}^{n}{\left(y_{i}-\bar{y}\right)}^{2}}$$

The closer the goodness of fit $R^{2}$ is to 1, the better the fitting degree of the multiple regression model to the observed value.

The hypothesis test of regression model, that is, the test of F statistical value.

Whether there is a linear relationship between the dependent variable m + 1st Road section *y* and *M* independent variables shown in the model needs to be tested. If all ${\hat{\beta }}_{j}$ are very small, then the linear relationship between y and is not obvious, so the original hypothesis can be
$$H_{0}:\beta_{j}=0,j=1,2,\cdots ,m$$(15){Q=Σi=1nei2=Σi=1n(bi−b^i)2U=Σi=1n(bi−b¯i)2
when is $H_{0}$ established, there is *F* statistic:(16)$$F=\frac{U/m}{Q/\left(n-m-1\right)}∼F\left(m,n-m-1\right)$$

At significance level of α, if
(17)$$F_{1-\alpha /2}\left(m,n-m-1\right)<F<F_{\alpha /2}\left(m,n-m-1\right)$$

Then accept $H_{0}$, otherwise reject.

The hypothesis test of regression coefficient, that is the test of t statistical value. Among the test of F statistical value, when the original hypothesis is rejected, $\beta_{j}$ is not all 0, but some of them are likely to be equal to 0. Therefore, further research on (m+1) assumptions should be carried out:$$H_{0}^{\left(j\right)}:\beta_{j}=0,j=1,2,\cdots ,m$$

When $H_{0}^{\left(j\right)}$ holds:(18)$$t_{j}=\frac{{\hat{\beta }}_{j}/\sqrt{c_{jj}}}{\sqrt{Q/\left(n-m-1\right)}}∼t\left(m,n-m-1\right)$$
(19)$$\left|t_{j}\right|<t_{\alpha /2}\left(m,n-m-1\right)$$
where $c_{jj}$ is the element in row *j* and column *j* of ${\left(X^{t}X\right)}^{-1}$. For a given α, if (19) holds,

Then we should accept $H_{0}^{\left(j\right)}$, otherwise reject it.

#### 3.4.4. Combination of MLR and LSTM

The (*m* + 1)-th road section is predicted through m adjacent road sections. The traffic flow of m road sections predicted after training through LSTM is combined with the predicted parameters obtained through multiple linear regression model. The formula is as follows:(20)$${\hat{Y}}_{L_{m+1}}=\beta^{T}\hat{L}$$

### 3.5. Model Prediction and Evaluation

Mean absolute error (MAE) and root mean square error (RMSE) are two important scales for evaluating the model in machine learning. Mae and RMSE are selected as the measurement indexes of the model in this study.
(21)$$\left\{ \begin{array}{c} \mathrm{RMSE}=\sqrt{\frac{1}{t}\overset{t}{\underset{t=1}{\Sigma }}{\left({\hat{Y}}^{t}-Y^{t}\right)}^{2}}\\ \mathrm{MAE}=\frac{1}{t}\overset{t}{\underset{t=1}{\Sigma }}\left|{\hat{Y}}^{t}-Y^{t}\right| \end{array}\right.$$
where, *t* represents the observation time corresponding to the traffic flow observation value in the test data set ${\hat{Y}}^{t}$, represents the traffic flow prediction value at time *t*, and $Y^{t}$ represents the actual observation value of traffic flow at time *t*.

## 4. Experimental Process

### 4.1. Experimental Platform and Environment

The experimental computer is configured with windows 10 64 bits operating system, Intel (R) core (TM) i5-8250u CPU and 8 GB memory. Programming language version is Python 3.8 (Guido van Rossum, Amsterdam, Netherlands), which is implemented in keras with tensorflow as the back end. Two kinds of software called MATLAB 2021 and SPSS 26 were used.

### 4.2. Parameter Setting

A two-layer LSTM model is created. At the same time, dropout function is added to prevent overfitting. We also use tanh activation function. Terminally MSE is used as the neural network of loss error. The parameters of the model are selected from empirical data and adjusted through many experiments. Finally, the sliding window length is 5, the number of hidden layer nodes is 80, the learning rate is 0.001, the batch size is 36, and the epoch is 200, as can be seen in Table 1.

### 4.3. Data Sources

Data comes from UK highway data collection website (http://tris.highwaysengland.co.uk/detail/trafficflowdata, accessed on 17 September 2022), map of expressway observation points in Britain (https://webtris.highwaysenglend.co.uk/, accessed on 17 September 2022) and the UK road map on Google Maps. The target section is in northern Waterford in southern Ireland. To be exact, it is around the place where longitude is 51.716354 and latitude is −0.385198.

Figure 4 shows the expressway map of some areas, and Figure 5 shows the traffic situation at the target intersection.

We display the data types used in this article, as shown in Table 2.

We select traffic flow data monitored every 15 min of each node in the sections of British place named as M25 j20-j21A, M1 j6-j6A, A405, M25 j21A-j22 and M1 j6A-j8, in which both of M and A is the prefix of the highway and j is the prefix of the mark on the different sections on one road. Data collected is the data of the whole month in July 2021, in which the interval of recording traffic flow is 15min. The selected road sections are represented by numbers in the figure, which are 1, 2, 3, 4 and 5 respectively, in which the value of m is 4 and Section 5 is the target section to be predicted.

### 4.4. Model Test of Multiple Linear Regression

The traffic flow data of four sections numbered 1 to 4 were selected at an interval of 15 min. Each road section selects 1000 data for multiple linear regression, i.e., M = 4, *n* = 1000. Order α=0.05.

Among the parameters in Table 3, $R^{2}$ is 0.8375, indicating that the fitting effect is good. The F statistic obtained by SPSS software is 1710.742F > $F_{\alpha /2}\left(m,n-m-1\right)\approx 2.38$, so it passed the *F* test.

Calculated by SPSS in Table 4, the five values of statistic T are 4.987, 15.920, 6.820, −2.293 and −0.259. For the inequality $|t_{j}|<t_{\alpha /2}\left(995\right)=1.96$, the original hypothesis cannot be rejected, which is not significant in the model. Therefore, we round off the fifth *t* value, that is, the fourth independent variable, and perform multiple regression analysis again.

Then we obtain Table 5 This time, the traffic flow of 1, 2 and 3 intersections is selected as the independent variable. The statistical values of T are 5.077, 16.028, 7.011, −3.146, respectively. In addition, $|t_{j}|<t_{\alpha /2}\left(995\right)=1.96$, so the original hypothesis can be rejected. Therefore, the *t*-test is completed.

It can be seen from the residual diagram in Figure 6 that the fitting effect is good.

We can obtain β^=[14.315,0.352,0.125,−0.024] to use in the MLR-LSTM model below.

## 5. Results

Figure 7 shows the pattern of the real data at 15 min intervals on the road section numbered 1. From it, it suggests that the data of traffic flow is periodic and unstable.

Firstly, we use LSTM to predict the traffic flow data of the sections named 1, 2, 3, 4, and 5, respectively, with time granularity of 15 min, 30 min and 60 min. We take the first 25 days of one month’s data as the training set and the next 5 days as the test set for prediction. After the network, we use MLR to predict each section by adjacent sections. Figure 8 shows the prediction error of the No.1 section with LSTM and MLR-LSTM. As can be seen from the perspective of RMSE and MAE, the prediction accuracy has been improved to some extent, and the accuracy improvement is the most obvious under the time granularity of 15 min.

We can conclude from Figure 8 that the improvement will be greater if the data with the granularity is shorter than 15 min.

In order to further verify the effectiveness of the model, we compared the results of the prediction method with those of various baseline methods. We select the traffic flow data of five adjacent intersections at 15 min time granularity, and calculate them using the methods proposed in this paper, Historical Average (HA), Arima, Gradient Boosting Regression Tree (GBRT), Support Vector Regression (SVR), Feed-forward Neural Network (FNN), Recurrent Neural Network (RNN), and Long Short-Term Memory(LSTM)*,* Gate Recurrent Unit (GRU) to obtain RMSE to evaluate the effect.

We can see from Table 6 that the method of using a neural network on this data set is better than the traditional machine learning method, and the effect is not very different among the three neural networks of RNN, GRU, and LSTM. The results will be more stable when using a neural network in the case of large data intervals.

Since the amount of data is about thousands, the time spent in building the network is not much. Taking an LSTM network as an example, the total training time is 48.586 s, and the training time of other neural networks is about this time.

Further, we predict the traffic flow data at each intersection under the granularity of 15 min by means of LSTM and MLR-LSTM, so as to obtain the universality of the model. The result is shown in Table 7. To conclude, the method we proposed has about a 14.83% performance improvement under the evaluation by RMSE and about a 16.51% performance improvement under the evaluation by MAE, as can be seen in Table 7.

As Figure 9 shows, we display four experiments based on MLR-LSTM by four sections. We can see it more clearly from the No. 5 section in Figure 10.

The curve of our model is smoother and the curve oscillation is reduced compared with the one based on LSTM. Intuitively, our prediction curve is closer to the original curve than the curve based on LSTM.

## 6. Conclusions

Motivated by the application of neural networks in the traffic field, this study presented a multi-methodological approach to forecast traffic flow. Combining multiple linear regression and long-term and short-term neural networks, our model has strong robustness, and high precision prediction results are obtained by our model.

This work is the first to establish the model based on MLR-LSTM to predict traffic flow data at multiple intersections. Through data experiment and empirical analysis, the combination of neural networks and statistical methods makes it possible to predict with high accuracy when the data is imperfect.

From a theoretical perspective, this study has established a comprehensive and novel prediction framework, which combines time and spatial data, and uses traditional statistical methods combined with neural networks of big data technology to build and improve the model. From a practical perspective, this study solves the problem of accurately predicting the future traffic flow with incomplete data.

In the experimental stage, the results of MLR-LSTM and LSTM are compared through the highway traffic flow data, and the prediction accuracy is improved by 14.83%. As well, the target link data to be predicted can still be accurately predicted in the case of partial missing data.

Our research has some limitations. On the one hand, it is about the limitations of data. The data in this paper are spatio-temporal data, but the factors that affect the traffic flow are not only spatio-temporal data, but also weather conditions [22], whether it is a holiday, the proportion of car models [23], and the individual behavior of drivers. In this study, no comprehensive analysis of other factors can be conducted, which will reduce the accuracy of prediction. On the other hand, it is the limitation of prediction methods. When selecting relevant roads, we only selected a small number of similar roads, and did not extract and analyze a large number of intersections through convolution network or consider multi-mode motion under real conditions [24]. Although the method in this paper simplifies the model and calculation, it may make some features not be captured and affect the final prediction results.

In future research, we should first focus on the impact of multiple factors on the traffic flow, and achieve multi-modal data fusion to predict the traffic flow. Random forest, gradient lifting tree, and other methods can be used for the comprehensive prediction of multiple factors. Secondly, in subsequent research, road feature extraction and convolution neural network [25] training in the whole local space can be considered to obtain better prediction accuracy. Finally, the time series data can be decomposed by Fourier transform or wavelet transform, so that the accuracy of the model can be higher.

## Figures and Tables

**Figure 1 sensors-22-07517-f001:**
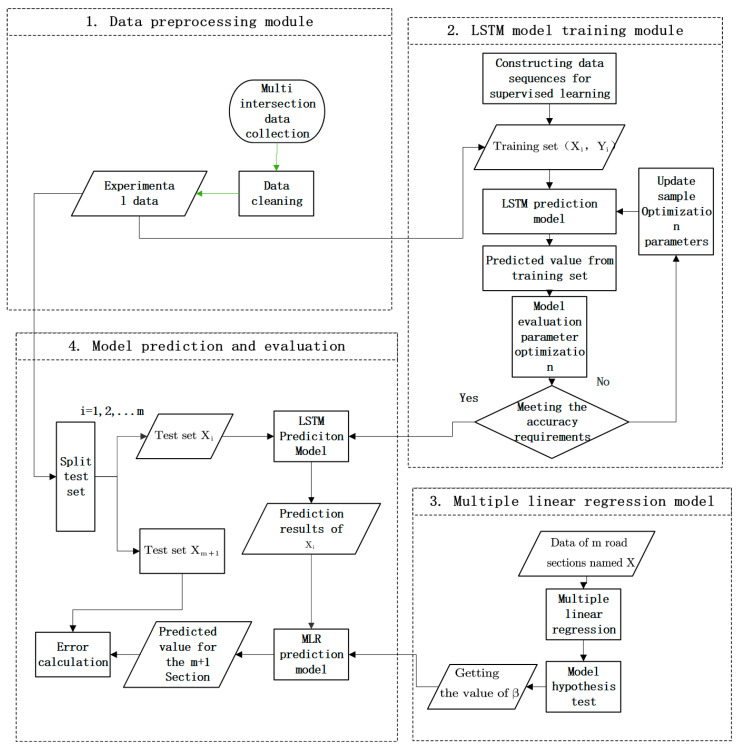
Framework of the proposed method for predicting traffic flow at multiple intersections.

**Figure 2 sensors-22-07517-f002:**
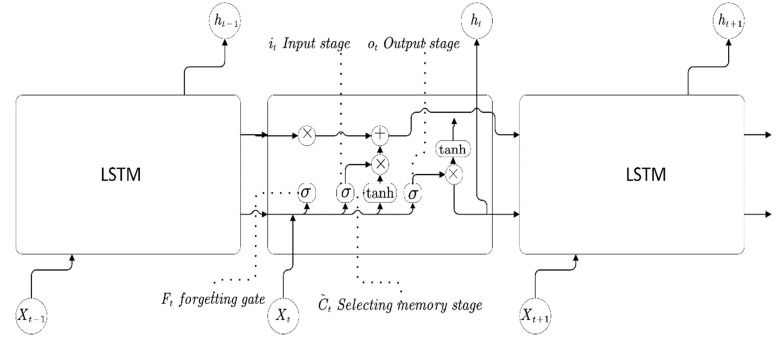
Internal structure diagram of the long-term and short-term memory mode.

**Figure 3 sensors-22-07517-f003:**
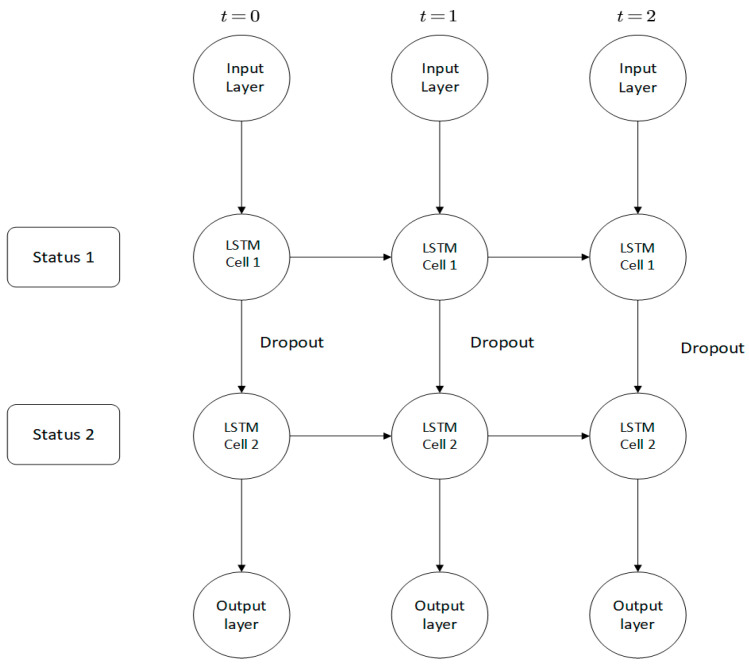
Adjustment process between LSTM cells.

**Figure 4 sensors-22-07517-f004:**
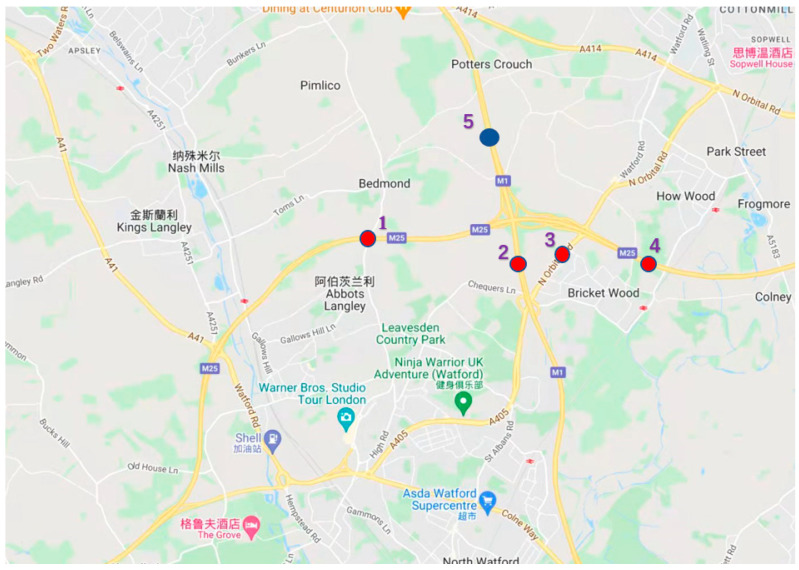
Map of the relevant intersection.

**Figure 5 sensors-22-07517-f005:**
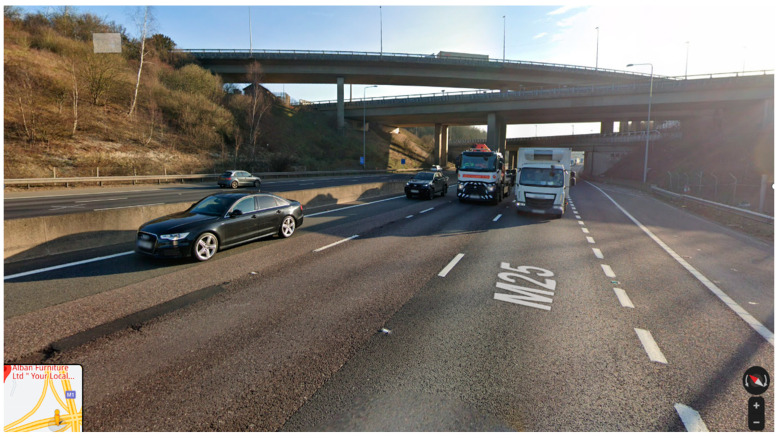
Real map of the traffic intersection from a western perspective on M25 highway.

**Figure 6 sensors-22-07517-f006:**
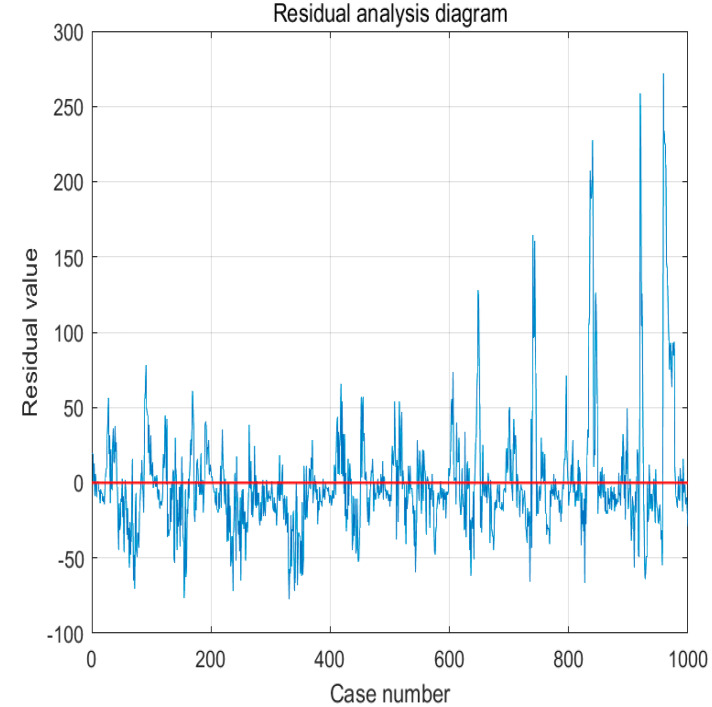
Residual analysis diagram.

**Figure 7 sensors-22-07517-f007:**
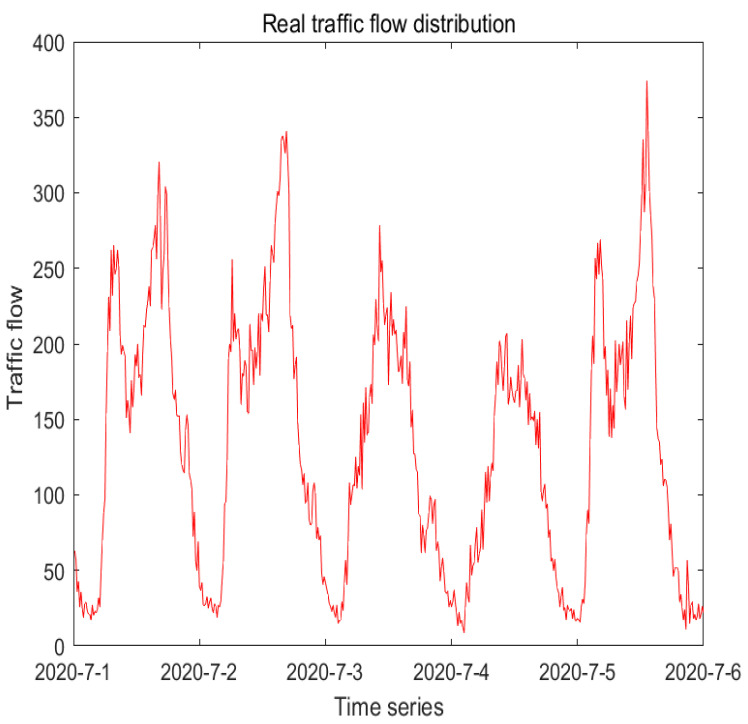
Display of real traffic flow sequence.

**Figure 8 sensors-22-07517-f008:**
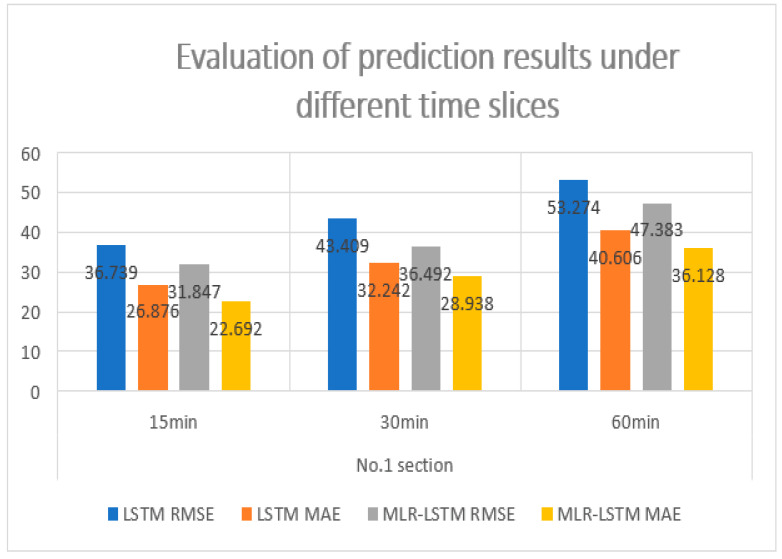
Evaluation of prediction results under different time slices.

**Figure 9 sensors-22-07517-f009:**
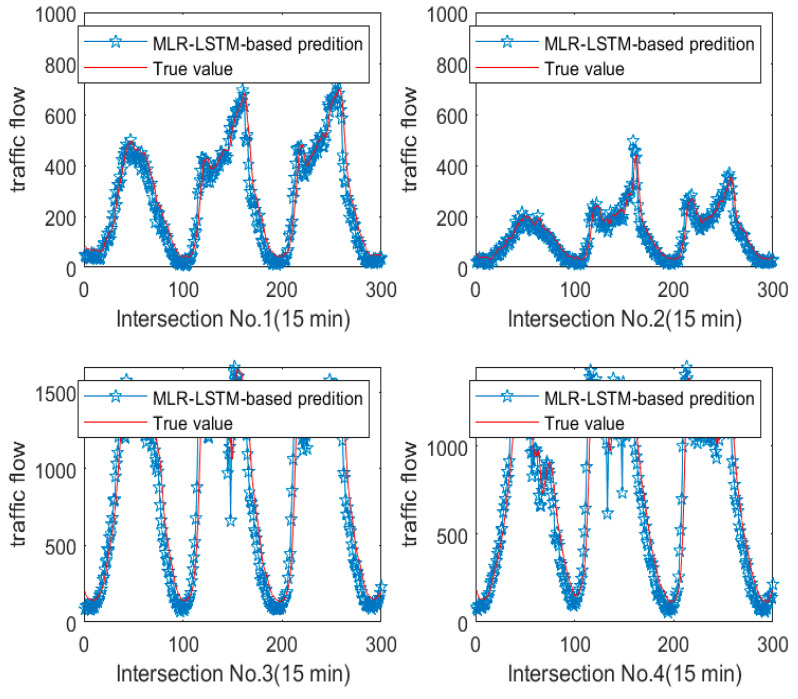
Prediction of traffic flow based on MLR-LSTM.

**Figure 10 sensors-22-07517-f010:**
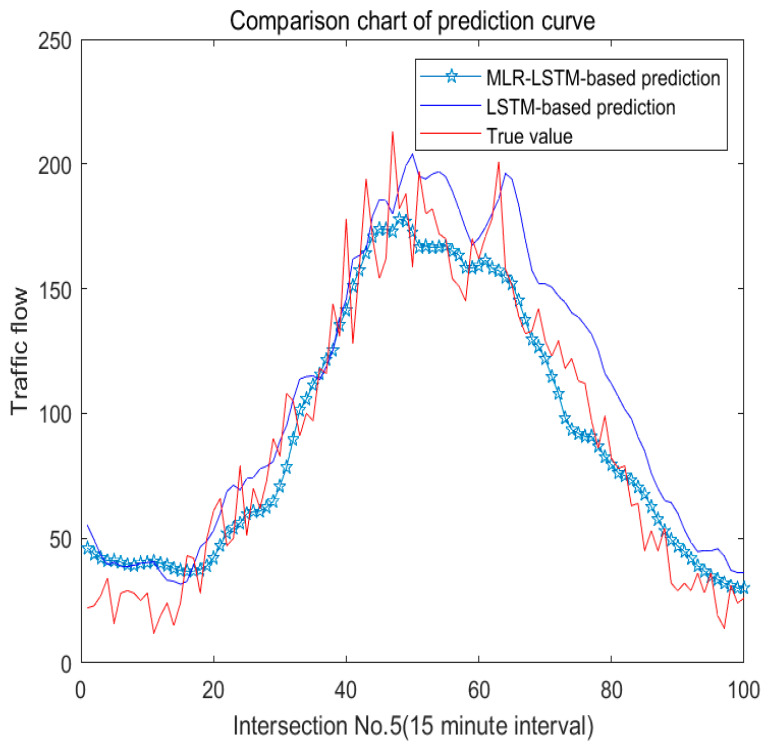
Comparison of prediction curves in a clearer way.

**Table 1 sensors-22-07517-t001:** Internal parameters of LSTM.

Parameter	Parameter Size	Explanation
Batch size	36	Number of samples per training
Epoch	200	Maximum number of rounds of training completed
Learning rate	0.001	Learning rate
sliding window length	5	the number of historical data

**Table 2 sensors-22-07517-t002:** Data types in the dataset.

Local Date	Local Time	Total Carriageway Flow		
		M25 j20-j21A	M1 j6-j6A	A405
2021-7-1	0:14:00	63	182	30
2021-7-1	0:29:00	57	138	30
2021-7-1	0:44:00	36	136	22
2021-7-1	0:59:00	43	112	14
2021-7-1	1:14:00	26	128	20

**Table 3 sensors-22-07517-t003:** *F* test statistics.

Model	Sum of Squares	Freedom	Mean Square	*F*	Significance
regression	7946372.914	3	2648791	1710.742	0.000
residual	1542135.262	996	1548.329		
total	9488508.176	999			

**Table 4 sensors-22-07517-t004:** Statistical value of the first *t*-test.

Model	B	Standard Error	Beta	*t*	Significance
(constant)	14.504	2.913		4.978	0.000
VAR00001	0.353	0.022	0.780	150.920	0.000
VAR00002	0.124	0.018	0.275	60.820	0.000
VAR00003	−0.022	0.010	−0.126	−20.293	0.022
VAR00004	−0.002	0.007	−0.009	−0.259	0.796

**Table 5 sensors-22-07517-t005:** Statistical value of the second *t*-test.

Model		B	Standard Error	Beta	*t*	Significance
1	(constant)	14.315	2.820		5.077	0.000
	VAR00001	0.352	0.022	0.778	16.028	0.000
	VAR00002	0.125	0.018	0.277	7.011	0.000
	VAR00003	−0.024	0.008	−0.135	−3.146	0.002

**Table 6 sensors-22-07517-t006:** Comparison of results from various baseline methods.

	RMSE					Average
	Section No. 1	Section No. 2	Section No. 3	Section No. 4	Section No. 5	
Proposed	31.847	29.035	19.352	68.392	81.394	46.004
HA	64.385	62.917	37.281	104.285	140.593	81.8922
Arima	62.388	58.482	33.185	96.284	132.592	76.5862
FNN	37.396	35.692	26.917	86.564	99.776	57.269
GBRT	43.271	41.302	27.984	84.776	111.885	61.8436
SVR	45.285	43.591	29.384	88.194	123.592	66.0092
LSTM	36.739	33.796	23.1	80.343	96.473	54.0902
GRU	35.383	32.943	24.938	81.834	95.184	54.0564
RNN	36.192	33.943	23.174	82.392	96.927	54.5256

**Table 7 sensors-22-07517-t007:** Summary of the performance between two methods.

Performance Measure	Section No. 1	Section No. 2	Section No. 3	Section No. 4	Section No. 5	Average Value
Method 1: LSTM-based method						
RSME	36.7391	33.796	23.100	80.343	96.473	54.091
MAE	26.87632	25.032	16.248	55.175	61.490	36.964
Method 2: MLR-LSTM-based method						
RSME	31.847	29.035	19.352	68.392	81.394	46.004
Improved percentage	13.31%	14.09%	16.23%	14.88%	15.63%	14.83%
MAE	22.692	20.748	13.449	46.341	51.259	30.898
Improved percentage	15.57%	17.12%	17.23%	16.01%	16.64%	16.51%

## Data Availability

The data used to support the findings of this study are available from the corresponding author upon request.

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
