# Peer review of "Multi-Section Traffic Flow Prediction Based on MLR-LSTM Neural Network"

_sensors, 2022, doi:10.3390/s22197517_

Round 1

Reviewer 1 Report

Multi-Section Traffic Flow Prediction Based on MLR-LSTM Neural Network

How the temporal information and spatial information of traffic flow data to transform traffic flow data from one dimension to multi dimension the authors should mention the details in clear manner.

preprocess data by data cleaning the authors failed to mention how it is done, techniques used.

The data is trained based on LSTM, in which the training set, test set and inspection set are set to train the parameters in the LSTM network, predict each adjacent section and evaluate the prediction results. What type of parameters they used for prediction.

Figure 1. Framework of the proposed method for predicting traffic flow at multiple intersections it is not clear and not meaningful

Figure 2. Internal structure diagram of the long-term and short-term memory mode it is not clear and not informative

The authors are writing the mathematical models which are not correct and no proofs for the solution

Reviewer 2 Report

This work focuses on traffic flow prediction using neural networks. A new approach based on linear regression and Long Short-Term Memory is proposed. Experimental results show superior performance of the proposed approach. The adopted approach is important for practitioners but this paper requires various modifications before it is published. 

1. The motivation of this work should be demonstrated more clearly. It is suggested to emphasize what is new compared with existing approaches.

2. The resolution of Figure 1 should be improve as it is the most crucial part in this work. In addition, more description should be added to the figure.

3. To show the importance of the research topic, more related works and the applications can be added to Section 1. For example: A competitive mechanism multi-objective particle swarm optimization algorithm and its application to signalized traffic problem.

4. The hyperlink in line 337 is invalid. Please fix it. 

5. There are some misprints in the manuscript. Please correct them. For example, it should be "4.3 Data sources" instead of "4.3Data sources".

6. The list of references should be carefully checked to ensure consistency with between all references and their compliance with the journal policy on referencing. For example, The page numbers are missing in Ref. [10].

Reviewer 3 Report

The paper proposes an urban traffic flow prediction method based on the combination of multiple linear regression and Long Short-Term Memory (MLR-LSTM). However, this paper still has some unclear expressions and some technical problems. Some comments that may help improve the quality of the paper.

(1) In section 1, the logic of stating the Introduction is not clear, and it is difficult for the reader to understand the current research status and challenges. In addition, the authors need to summarize the challenges and contributions of this study

(2) The depth and scope of section 2 (Related Works) are inadequate. Besides, the structure of this section is also not reasonable enough. The related work is not only the introduction of the methods, but also show the readers the main background of this research field. So that the readers can easier to follow the manuscript and comprehend the motivation of it.

(3) The authors need to clearly state the definition of the loss function.

(4) The depth of section 5 (Results) are inadequate. MLR-LSTM needs to be compared with existing SOAT baseline methods to reflect the advantages of the MLR-LSTM model. If possible, the authors may consider providing information on training and inference time for reference, which is also an important performance index, considering that recurrent networks are relatively harder to train.

(5) Typesetting issues hindered the readability of this manuscript. I think a manuscript with professional typesetting is respect for research. In addition, there are some typos and mistakes, the authors should polish their paper carefully.

(6) Authors should refer to recent references, e.g.

[1] Xinyu Chen, et al. Bayesian Temporal Factorization for Multidimensional Time Series Prediction[J]. IEEE Transactions on Pattern Analysis and Machine Intelligence.2022. 44(9): 4659-4673.

[2] Peixiao Wang, et al. A Multi-view Bidirectional Spatiotemporal Graph Network for Urban Traffic Flow Imputation[J]. International Journal of Geographical Information Science. 2022. 36(6):1231-1257

[3] Shaokun Zhang, et al. A Graph-Based Temporal Attention Framework for Multi-Sensor Traffic Flow Forecasting[J]. IEEE Transactions on Intelligent Transportation Systems. 2022. 23(7): 7743 - 7758

Round 2

Reviewer 1 Report

Paper can be accepted 

Author Response

Thank you for your affirmation of my manuscript.

Reviewer 3 Report

I have no further comments.

Author Response

(The authors gave the same response as above.)
